# Hit Compounds and Associated Targets in Intracellular *Mycobacterium tuberculosis*

**DOI:** 10.3390/molecules27144446

**Published:** 2022-07-12

**Authors:** Clement K. M. Tsui, Flavia Sorrentino, Gagandeep Narula, Alfonso Mendoza-Losana, Ruben Gonzalez del Rio, Esther Pérez Herrán, Abraham Lopez, Adama Bojang, Xingji Zheng, Modesto Jesus Remuiñán-Blanco, Yossef Av-Gay

**Affiliations:** 1Department of Medicine and Microbiology and Immunology, Life Science Institute, University of British Columbia, Vancouver, BC V6T 1Z3, Canada; ctsui@mail.ubc.ca (C.K.M.T.); flaviasorrentino@gmail.com (F.S.); gagandeepnarula@gmail.com (G.N.); abraham.x.lopez@gsk.com (A.L.); albojang@yahoo.com (A.B.); xingji.zheng@ubc.ca (X.Z.); 2National Centre for Infectious Diseases, Tan Tock Seng Hospital, Singapore 308442, Singapore; 3GSK, Global Health Medicines R&D, PTM, Tres Cantos, 28760 Madrid, Spain; almendoz@ing.uc3m.es (A.M.-L.); ruben.r.gonzalez-del-rio@gsk.com (R.G.d.R.); esther.x.perez-herran@gsk.com (E.P.H.); modesto.b.remuinan@gsk.com (M.J.R.-B.); 4Department of Bioengineering and Aerospace Engineering, Carlos III University of Madrid, 28040 Madrid, Spain

**Keywords:** sensitivity, antimicrobial resistance, selection, screening, chemical genetics, bioinformatics, drug discovery, tuberculosis

## Abstract

*Mycobacterium tuberculosis* (*Mtb*), the etiological agent of tuberculosis, is one of the most devastating infectious agents in the world. Chemical-genetic characterization through in vitro evolution combined with whole genome sequencing analysis was used identify novel drug targets and drug resistance genes in *Mtb* associated with its intracellular growth in human macrophages. We performed a genome analysis of 53 *Mtb* mutants resistant to 15 different hit compounds. We found nonsynonymous mutations/indels in 30 genes that may be associated with drug resistance acquisitions. Beyond confirming previously identified drug resistance mechanisms such as *rpoB* and lead targets reported in novel anti-tuberculosis drug screenings such as *mmpL3*, *ethA*, and *mbtA*, we have discovered several unrecognized candidate drug targets including *prrB*. The exploration of the *Mtb* chemical mutant genomes could help novel drug discovery and the structural biology of compounds and associated mechanisms of action relevant to tuberculosis treatment.

## 1. Introduction

*Mycobacterium tuberculosis* (*Mtb*), the etiological agent of tuberculosis (TB), is one of the most devastating infectious agents in the world [1]. In 2020, about 1.5 million people died from the disease, and 10 million people developed the illness (http://www.who.int/en/news-room/fact-sheets/detail/tuberculosis, accessed on 15 June 2022). TB transmission is airborne, where droplets containing *Mtb* enter the lungs and circulating alveolar macrophages engulf the bacilli (http://www.who.int/mediacentre/factsheets/fs104/en/, accessed on 15 June 2022). Macrophages are key components of the human innate immune system that destroy invading microorganisms. However, *Mtb* can survive and persist from the macrophage’s killing machinery and even replicate inside the macrophage in a specified organelle termed the phagosome. *Mtb* can evade the host immune system and is protected from many antibiotics that fail to reach the phagosome [2].

There are several first- and second-line anti-TB drugs, and the treatment involves a regime of four drugs, isoniazid, rifampin, pyrazinamide, and ethambutol taken daily for six to nine months, a far longer treatment than for most bacterial infections. With the increasing prevalence of multi- and extremely drug-resistant TB, treatment of patients often involves the use of more expensive second-line drugs and requires over 24 months. A few candidate drugs and hit compounds have been discovered in the last two decades, but only two drugs, bedaquiline (BDQ) and pretomanid, have been FDA-approved in the past 40 years [3], plus there is an urgent need to combat these new *Mtb*-resistant strains by refueling the drug development pipeline with novel drug discovery approaches.

One of the challenges in TB drug discovery is the lack of successful transfer from compounds with in vitro activity to efficacy in the clinical settings. For instance, compounds may be selected that are only active in vitro conditions; the targets which are inhibited and identified in in vitro liquid culture during cell-based phenotypic screens may not be essential in vivo [4]. This can be achieved through a more global understanding of the host–*Mtb* interaction using a chemical-genetic approach. We have developed an advanced intracellular drug-screening assay to screen compounds in infected macrophages [5,6]. Using our approach, we have further screened two libraries and identified a set of diverse chemical entities that are highly effective against *Mtb* within the human macrophage with marked intracellular selectivity. However, the mode of action (MOA) in these hit compounds and the potential drug targets/inhibitors have not been fully elucidated.

Whole genome sequencing (WGS) technology followed by bioinformatics analysis has been effective to investigate the epidemiology and transmission of *Mtb* in outbreak investigation and for infection control [1,7,8,9]. The technology and variant calling pipeline are also useful in characterizing the genetic polymorphisms and mechanism of resistance in drug resistance clinical strains [10,11,12,13,14]. Drug resistance in *Mtb* is mainly conferred by single nucleotide polymorphisms (SNPs) and indels in genes encoding drug targets or drug-converting enzymes [11,12]. In addition, metabolic/physiological changes associated with drug tolerance such as changes in efflux pump regulation and those in response to the host immune responses may have an impact on the emergence of drug resistance [14]. Since the lack of understanding of the compound MOA has become a major barrier to the development of potential novel drug to TB treatment, the WGS technology has been extended to identify and characterize the candidate drug targets in novel TB drug discovery programs; the genomes of spontaneous drug-resistant mutants of *M. bovis* BCG or *M. smegmatis* from screening have been characterized followed by variants annotation as well as target identification and validation [15,16,17,18].

In this study, we identified potential drug targets and MOAs for selected hit compounds active intracellularly in THP-1 cells, based upon their chemical properties. We further generated and sequenced the genomes of 53 resistant mutants of *Mtb* H37Rv against various hit compounds to assist the identification of their corresponding MOAs. The identified mutations lead us to suggesting novel MOAs involving candidate proteins. These include possible drug targets which are critically important to identify new antibiotics for the long-term control of TB disease.

## 2. Results and Discussion

### 2.1. Identification of Hit Compounds

The previous screening of GSK-proprietary libraries has identified a set of diverse chemical entities. High-throughput screening (HTS) was performed using a 5 μM single shot and a 1–10 μM dose response for 84,000 compounds from predefined in-house GSK libraries; first, the “TB box” containing 11,000 compounds came out from in vitro phenotypic screening of 2,000,000 compounds against *M. bovis* BCG with hit confirmation in *Mtb*, and second, a library of 73,000 compounds was drawn from GSKChem with “ideal” medicinal chemistry characteristics termed “Small&Beautiful”.

The data were analyzed, and the intracellular MIC50 and MIC90 were extrapolated for all compounds tested. Five hundred twenty-three hit compounds belonged to the GSK “TB box”, with an intracellular MIC90 of <3 μM, and 31 hit compounds belonged to the “Small&Beautiful”, with an MIC90 of <10 μM. A total of 564 hits from the HTS campaign were identified in both aforementioned libraries.

After the removal of duplicates, as well as compounds with known MOAs and compounds cytotoxic in HepG2 cells, 265 hit compounds that were active at low micro-molar concentration were identified. Most of these hits were more active intracellularly, and 85 compounds (32%) were demonstrated to be active only in an intracellular assay (unpublished data). Then, we performed structure-based hit clustering and classified these 85 hit compounds to known chemical structures, which were similar to 19 previously identified compounds with either known activity, MOA, or targets in *Mtb* (Figure 1). These included 18 compounds that are dihydrofolate reductase (DHFR)-like inhibitors, 11 inosine 5′-monophosphate (IMP), 10 *N*-benzyl-6′,7′-dihydrospiro[piperidine-4,4′-thieno [3,2-c]pyrans] (SPIRO), 10 tetrahydropyrazolo[1,5-a]pyrimidine-3-carboxamides (THPP), and eight compounds that were previous identified as mmpL3 inhibitors; other targets included *Mtb* gyrase inhibitors (MGIs) and oxazolidinone. Based on the MOA or target in *Mtb*, 12 of these chemical entities were targeting DNA gyrase and 11 targeting IMP (Figure 1).

Rationalizing that it would be difficult to obtain mutants for intracellular *Mtb*, we mimicked the intracellular environment within the macrophages by performing in vitro MIC in various carbon sources; the 85 hit compounds selected were subject to in vitro MIC in different carbon sources (glucose, cholesterol, and acetate) of which 27 (34%) showed activity in the MIC90 assay in cholesterol. Most of the compounds active in cholesterol were also active in acetate.

### 2.2. Generation of Mutants, Mutant Characterization, and WGS

Out of these 85 hit compounds, the properties of 16 compounds were summarized in Table 1 and Figure 2. These compounds demonstrated potent activities against *Mtb* in defined carbon source media [5,6]. Although some compounds were anticipated to have antitubercular activity based on the structural analysis, the targets/MOAs of some compounds were unknown or undefined.

Drug resistance in *Mtb* is always caused by mutations in existing genes that are inherited—passed from the parental (mother) to mutants (daughter cells). Following phenotypic screening, about 3–6 resistant colonies were obtained for each compound. One to six resistant mutants were selected for WGS, which allowed a genetic polymorphism comparison among different colonies. The laboratory parental strain H37Rv was also sequenced to exclude SNPs derived from in-house passage in comparison to the standard reference genome (NC_000962.3). For each mutant, 151,782–2,071,066 reads were generated; the depth of coverage varied among samples and ranged from 14 to 186 (average coverage = 65×) (Appendix A). Over 95% of the trimmed reads were mapped to the reference genome in NCBI showing a very low level of contaminants (Appendix A). Two mutants 648X-1 and 648X-2 contained contaminated reads which were excluded before mapping to the reference genome. The size of the *Mtb*-assembled genome was about 4.4 Mb in agreement with the reference genome size.

### 2.3. Identification of Candidate Intracellular Drug Targets in Resistant Mutants

We discovered 74 nonsynonymous SNPs and 13 indels (insertions or deletions) located in 30 different genes that arose in 53 *Mtb* mutant genomes (Table 2 and Appendix A). Most polymorphisms (28) localized in genes involved in cell walls and cell processes, followed by intermediary metabolism and respiration (16) and regulatory proteins (14) (Figure 3).

The annotation of these mutations revealed several known genes or targets that are relevant to drug resistance; several mutations were reported in targets of the first- or second-line drugs in treating TB. We found a mutation in *rpoB* (Asp574Asn) encoding the beta subunit of RNA polymerase in mutants to compound 950A. Mutations in *rpoB* known to confer resistance to rifampicin are commonly reported in MDR and XDR *Mtb* strains [8,10]. Alternatively, six resistant strains to three compounds (472A, 739A, and 912A) were found to possess mutations (three different nonsynonymous mutations) in *ethA* (Table 2), an FAD-containing monooxygenase, which is a mycobacterial enzyme responsible for bio-activation of ethionamide (ETH), an antibiotic prodrug in TB treatment [19,20,21,22]. Loss-of-function mutations in *ethA* result in ETH resistance [20]. Interestingly for four mutants resistant to 472A and 739A and bearing substitutions in *ethA*, they also possessed indels in *Rv3220c*, which belongs to a two-component regulatory system that enable the organisms to make coordinated changes in gene expression in response to environmental stimuli [23]. However, *Rv3220c* did not appear to contribute to *Mtb* virulence in a mice model [23]. For the two mutants resistant to 472A and 739A but without mutations/indels in *ethA* and *Rv3220c*, mutations were observed in *rpsO*, which is important in protein translation [24].

Apart from *ethA*, mutants resistant to 912A and 705A had mutations or gained a stop codon in *Rv3083* (*MymA*), which also plays a role in activating ETH; the loss of *MymA* function resulted in ETH-resistant *Mtb* [25,26]. Grant et al. (2016) found that *MymA*, a Baeyer−Villiger monooxygenases (BVMO) not previously described as an activating enzyme, is required to oxidize compounds to the corresponding sulfoxide for its replicating and non-replicating activity [25]; the loss of *MymA* function is proposed to confer resistance comparable to the loss of *ethA* function [26].

Gene targets with independent mutations may confer a fitness advantage to *Mtb* strains in the presence of antimicrobial drugs. Importantly, we identified *mmpL3* as a target of independent mutation in mutants resistant to compounds 267A, 213A, and 290A. *mmpL3* is a membrane transporter in the resistance-nodulation-cell division family and has been shown to be the target of several small molecules and antimycobacterial compounds [27,28,29,30]. Similarly, multiple strains resistant to 412A developed mutations in *prrB*, which belongs to a two-component regulatory system composed of *prrB* histidine kinase and *prrA* response regulator [31,32,33]. This gene has been shown to be critical for the viability of *Mtb* cells and is required for the initial phase of macrophage infection. The *prrB* gene was also reported as the target of a hit compound diarylthiazole [32].

In addition, mutations were reported in genes associated with BDQ resistance. A frameshift insertion leading to loss-of-function in Rv0678 has been observed in mutants resistant to 454A. *Rv0678* is a gene that regulates the expression of the MmpS5-MmpL5 efflux pump, of which the variants could confer resistance to BDQ, leading to 2- to 8-fold increases in BDQ MIC as well as 2- to 4-fold increases in clofazimine MIC [34]. They have been isolated in vitro upon exposure to clofazimine or BDQ. Interestingly, a cross-resistance between clofazimine (CFZ) and BDQ was shown to be due to mutations within *Rv0678* [34,35,36,37], a transcriptional repressor, which results in the derepression and upregulation of the multi-substrate efflux pump *mmpL5*. Similar genetic polymorphisms (SNPs and indels) in *Rv0678* have also been reported in resistant mutants to “compound 5” in a drug target discovery study [38]. Interestingly, all the mutants bearing indels in *Rv0678* also had deletion in *mbtA*, which is an adenylating enzyme that catalyzes the first step in the biosynthesis of the mycobactins [39]. In the last decade, the siderophore biosynthesis has been pursued as a drug target in TB [40]. The analog, 5′-O[N-(salicyl)sulfamoyl]adenosine was shown independently by three groups to inhibit *MbtA* with a Ki value of ∼6 nm [41,42].

Our study also revealed mutations in genes that are not well investigated or unknown in function. For instance, independent mutations were observed in TB18.5, a conserved hypothetical protein without known function, in mutants to 296A. TB18.5 has been predicted to be an outer membrane protein (https://mycobrowser.epfl.ch/, accessed on 15 August 2021) and could be useful in clinical pathogen diagnosis [43]. Other candidate targets include narL discovered in mutants to 412A. NarL belongs to one of the two-component regulatory systems and regulates the synthesis of formate dehydrogenase-N and nitrate reductase enzymes during aerobic nitrate metabolism [44]. Lead compound targeting NarL is being explored for *Mtb* treatment [45]. Mutations were observed in polyketide synthase *pks6*, in two mutants (to compound 296A) having mutations in TB18.5. *Pks6* is involved in human infection [46]. In addition, substitution was reported in CtpC from the mutant to 648X; *ctpC* appeared to be important in the transport of heavy metal zine and contributed to the survival of *Mtb* in macrophage [47,48]. In addition, mutants from compound 486X were found to have mutations in *phoR* and *fbiA*/*fbiC*, which have important functions. PhoPR is a well-known two-component regulator of pathogenic phenotypes, including the secretion of the virulence factor ESAT-6, the biosynthesis of acyltrehalose-based lipids, and the modulation of antigen export [49,50,51]. Clinical mutants resistant to delamanid, a drug against *Mtb*, were found to possess mutations in *fbiA* and *fbiC*, as well as *fbiB* [52]. Likewise, mutations in *fbiA* and *fbiC* have been related to resistance to delamanid in *M. bovis* BCG mutants.

Mutants to 622A and 1114A have mutations in *virS* which is important—the expression of *mymA* operon genes may be regulated through PknK-mediated phosphorylation of VirS [53]. *VirS* is important for *Mtb* to block the phagosomal−lysosomal fusion in the activated macrophages and to survive in acidic conditions [54]. Another mutant to compound 1114A has mutations in *dnaE1*, which is essential for high-fidelity DNA replication and is considered a potential drug target [55,56]. A mutant to 412A has a mutation in isoniazid inducible gene *iniB*, which is involved in cell wall synthesis [57], while *moaC3* (mutant to 412A) is part of the Molybdenum cofactor (Moco) biosynthesis pathway, which may be significant to pathogenesis [58].

Multiple mutants to 213A, 622A, and 1114A gained the N-terminal stop codons in *sugI*, which encodes a sugar-transport membrane protein in *Mtb* [59]. The same mutation may be associated with the resistance to the second-line drug D-cycloserine; Chen et al. [13] suggested the loss-of-function mutation discovered from a mutant may result in a lower uptake of cycloserine inside the cell, therefore leading to higher resistance to d-cycloserine.

The role of mutations in the genes of unknown functions or “relatively low-abundance” genes such as *Rv0370c*, *Rv3629*, *Rv1948*, *Rv1825*, *Rv0585c*, *Rv3175*, and *Rv3327* is unclear; these mutations may be random or involved in compensating for resistance mutations or providing an additional level of resistance [60]. Fitness costs caused by chemical resistance mutations could be ameliorated by compensatory mutations, which do not contribute directly to drug resistance [60]. In fact, the WGS of MDR and XDR strains also revealed lots of mutations, and some of them may be trade-off or involved in compensation of fitness costs [10].

## 3. Materials and Methods

### 3.1. Preparation of Chemical Compounds

Hit compounds with potential anti-tuberculous activities have been identified, and the chemicals were prepared under similar conditions as previously described in Sorrentino et al. [6].

### 3.2. Libraries

The “TB box” library is a collection of 11,000 compounds that have shown a biological effect (DR curve) from any of the phenotypic HTS campaigns run against *Mtb* and *M. bovis*. Compounds with structures related to MOAs known as antitubercular have been removed. Compounds showing pTOX50 values higher than 6.3 (<0.5 μM) have been removed.

The “Small&Beautiful” library is composed of compounds drawn from GSKchem, filtered on size and lipophilicity, 10 ≤ heavy atom count (HAC) ≤ 28, and −2 ≤ ClogP ≤ 3, filtered on “promiscuity” (multiple targets, side effects, and unsuitable DMPK) and inhibition frequency index (IFI) ≤ 3%, filtered using other physicochemical properties, i.e., MW ≤ 400, RotBonds ≤ 5, 0 ≤ HBD < 8, 0 ≤ HBA ≤ 40 ≤ Neg ≤ 2, 0 ≤ Pos ≤ 2, AromRings ≤ 2, and TotRings ≤ 3, as well as filtered on “shapeliness” (roundness) and fCsp3 ≥ 0.3 (i.e., ≥30% of carbon atoms must be sp3), and filtered on ALS availability with 150 µL as the minimum in UP ALS required. Diversity selection and redundancy elimination were also performed.

### 3.3. HTS

We utilized an ex vivo HTS assay to test compounds activity against intracellular *Mtb*. Initially, we used the raw-intensity luciferase method as described [5,6]. Briefly, *Mtb* cultures were opsonized and used to infect the THP-1 cells at a 1:2 multiplicity of infection (MOI). Three hours following infection, the medium was aspirated and replaced with a medium containing the compounds. After 24, 48, or 72 h, infected macrophages were harvested and lysed. Bright-Glo reagent (Promega TM052) was added to each culture, and luciferase activity was measured with an aTropix TR7171 luminometer (Applied Biosystems, Foster City, CA, USA) available in the BCL-3 facility. The process was repeated with fractionated and purified peptide, until a pure homogenized compound was identified.

### 3.4. Bacterial Culture and Mutant Conditions

*Mtb* H37Rv (ATCC 25618) was utilized for all experiments. *Mtb* strains (WT and all chemical mutants) were grown in 7H9 broth (Difco) supplemented with albumin-dextrose-catalase enrichment (ADC), 0.05% Tween 80. *Mtb* H37Rv was grown to the mid-logarithmic phase in a 7H9 broth supplemented with ADC (10%) and Tween 80 (0.05%). OD_600_ was measured, and the bacteria was spun and resuspended in media to a final concentration of ~1 × 10^8^ CFU/mL. The entire volume was plated onto Middlebrook 7H10 agar supplemented with 10% ADC 0.05% Tween 80, 0.2% glycerol, a carbon source of interest, and a 2× MIC90 compound concentration. Bacteria were also plated onto compound-free plates as controls. Plates were incubated for 4–6 weeks at 37 °C containing 5% CO_2_ or until colonies were observed. The spontaneous rate of resistance was calculated as the number of colonies on compound-containing plates divided by the total number of viable bacteria estimated on compound-free plates. Isolated resistant colonies were picked from compound-containing plates and replated onto fresh 2× MIC90 compound-containing Middlebrook 7H10 plates for the confirmation of resistance. Once confirmed, colonies were picked and grown in a 7H9 broth supplemented with 10% ADC and a 2× MIC90 compound concentration and grown to the mid-logarithmic phase. Genomic DNA was extracted using the lysozyme method [61].

### 3.5. WGS of Mutants

Wild-type *Mtb* H37Rv and mutants were characterized by WGS as described in [62]. Briefly, the DNA libraries were constructed with a Nextera XT DNA kit (Illumina, San Diego, CA, USA). The DNA was fragmented and purified with AMPure XP beads. DNA libraries of wild-type H37Rv and mutant samples were normalized and sequenced using the MiSeq platform with 2 × 250 cycles (MiSeq Reagent Kit v2) at British Columbia Center for Disease Control Public Health Laboratory and at Genome Québec Innovation Centre at McGill University.

### 3.6. Bioinformatics Analysis

The quality of the reads was assessed by Fastqc (http://www.bioinformatics.babraham.ac.uk/projects/fastqc/). Reads were quality-trimmed by Trim Galore (http://www.bioinformatics.babraham.ac.uk/projects/trim_galore/). Trimmed sequence reads were aligned to the reference genome sequence of H37Rv (NC_000962.3) using BWA-mem [63], and SNVs and indels were called using GATK v.3 [64]. The SNVs and indels generated using GATK were filtered to ensure high confidence. The parameters for filtering were as following: (i) minimum read depth of 10; (ii) maximum base quality of 30 for every nucleotide in the sample; (iii) minimum mapping quality of 20. Variants of phred-scaled base quality scores above 100 were selected. SnpEff [65] was used to annotate and to output the SNVs changes in mutants according to the reference genome and GFF files of *Mtb* H37Rv in NCBI. Unique variants in mutants were identified by examining the discordant SNVs between the wild type and mutants that differed from the H37Rv reference in NCBI [66]. To avoid false-positive SNVs, the unique variant in mutants was inspected through a tablet [67]. SNPs and indels occurring in PPE/PE_PGRS genes, which contained repetitive elements, were excluded to avoid inaccuracies in the read mapping and alignment in those portions of the genome [10,38]. Mutations arising from the comparison between the parental strain *Mtb* H37Rv and the standard reference genome (NC_000962.3) were also excluded from the analysis. Reads from contamination data and other bacteria were excluded after analysis by Kraken v.1 [68]. Raw reads were also assembled using SPAdes v.3.9.0 [69] with *k-mer* sizes of 21, 33, 55, 77, and 99.

## 4. Conclusions

The bioinformatics analysis of 53 mutants screened against various compounds identified several promising genes that conferred resistance to given chemical entities and, as such, may provide novel drug targets. Some targets of these chemical libraries were consistent with those that have been tied to the proposed mechanism of action or resistance (e.g., *rpoB*, *mmpL3*, and *ethA*) and a potential new pathway identified in our analysis (e.g., *prrB*). The analysis has extended our understanding of the biological basis for the anti-tuberculous actions. Future studies are needed to address the role of the identified mutations in genes of unknown functions and how they might be involved in the MOA or resistance of these compounds to TB.

## Figures and Tables

**Figure 1 molecules-27-04446-f001:**
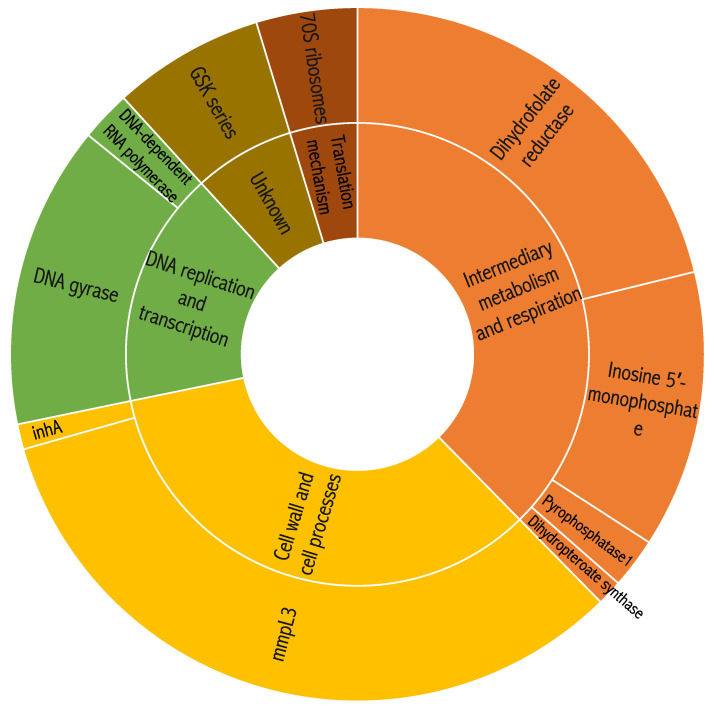
Clustering analysis of the modes of action (MOAs) from 85 compounds.

**Figure 2 molecules-27-04446-f002:**
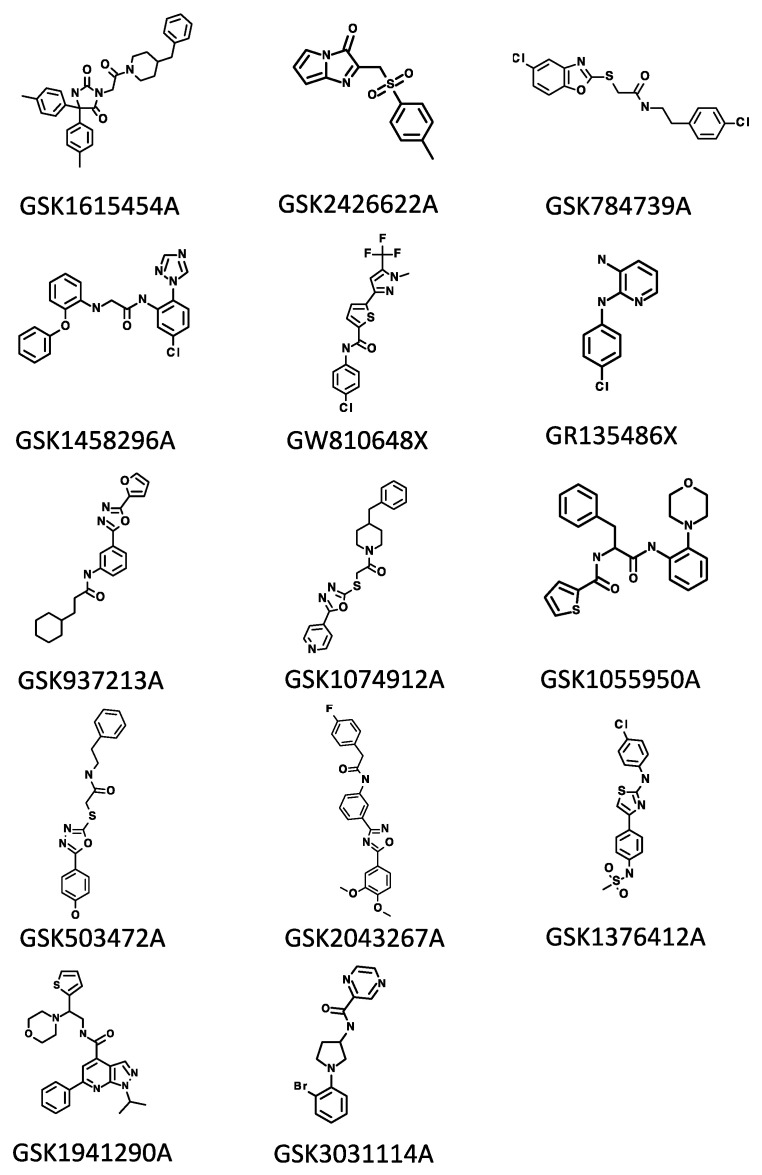
Structures of compounds. Formulas were converted to structures using the PubChem structure tool (https://pubchem.ncbi.nlm.nih.gov//edit3/index.html, accessed on 15 June 2022).

**Figure 3 molecules-27-04446-f003:**
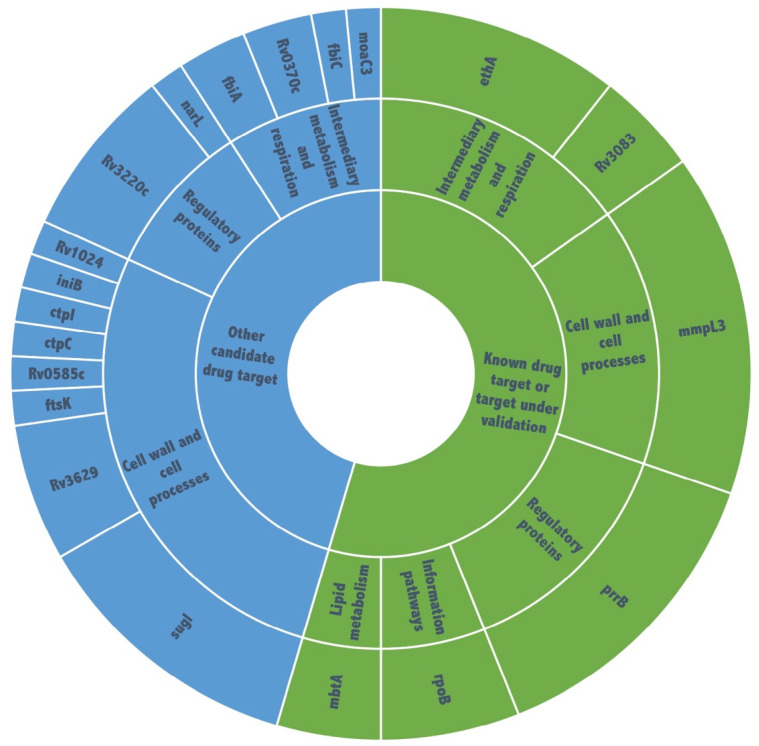
Function of candidate target genes identified from *Mycobacterium tuberculosis* H37Rv mutants.

**Table 1 molecules-27-04446-t001:** Characteristics of the chemical compounds and conditions for mutant screening.

ID	Molecular Weight	Intracellular MIC90 (μM)	Resistant Mutant Selection Media	Number of Resistant Colonies
213A	365	0.6	5× MIC glucose	5
267A	433	0.16	2× MIC90 ADC	3
290A	476	0.6	2× MIC90 ADS	2
950A	436	1.26	2× MIC90 ADC	4
739A	381	2	2× MIC90 acetate	3
472A	355	2	2× MIC90 acetate	4
412A	380	0.32	2× MIC90 ADC	3
412A	380	0.32	2× MIC90 glucose	6
296A	420	1	2× MIC90 acetate	6
648X	386	1.58	2× MIC90 ADC	2
454A	496	2	2× MIC90 ADC	3
1114A	347	2.51	5× MIC90 glucose	2
486X	220	0.16	5× MIC90 ADC	3
912A	394	1.58	2× MIC90 acetate	4
622A	288	2	5× MIC90 glucose	3
705A	310.8	1.26	2× MIC acetate	1

**Table 2 molecules-27-04446-t002:** List of SNPs and indels in various genes recovered from various mutants after whole genome sequencing (WGS). Gene names and features were displayed according to Mycobrowser annotations (https://mycobrowser.epfl.ch/, accessed on 25 August 2021).

Compound	Number of Mutants	Gene/ ORFs	Genetic Polymorphisms (Frequency)	Relevant Codon Change (Frequency)	Product
213A	5	*mmpL3*	755A > G; 758G > A; 875T > C; 1985C > A; 2051T > C	Tyr252Cys; Gly253Glu; Ile292Thr; Ala662Glu; Val684Ala	Conserved membrane transport protein
		*sug* *l*	16C > T	Gln6*, gained stop codon (2)	Involved in transport of sugar across the membrane. Translocation of the substrate.
267A	3	*mmpL3*	763T > C; 765C > G; 1932 C > A	Phe255Leu (2); Phe644Leu	Conserved membrane transport protein
290A	2	*mmpL3*	1909C > A	Leu637Ile (2)	Conserved membrane transport protein
		*Rv0370c*	474G > T	Val158Val (2)	Unknown. possible Oxidoreductase
950A	4	*rpoB*	1720G > A	Asp574Asn (4)	Transcription of DNA into RNA
		*Rv3629*	641G > A	Gly214Glu (4)	Probable conserved integral membrane protein
739A	3	*ethA*	611T > C	Met204Thr (2)	Monooxygenase that activates the pro-drug ethionamide (ETH)
		*rpsO*	157C > T	Arg53Trp	30S ribosomal protein S15
		*Rv1024*	154C > T	Pro52Ser	Possible conserved membrane protein
		*Rv3220c*	746C > CA (2)	indels, frameshift variant	Probable two-component sensor kinase
472A	4	*ethA*	611T > C	Met204Thr (3)	Monooxygenase that activates the pro-drug ETH
		*rpsO*	157C > T	Arg53Trp	30S ribosomal protein S15
		*Rv3220c*	746C > CA (3)	indels, frameshift variant	Probable two-component sensor kinase
412A	9	*prrB*	452T > C (7); 548C > T; 875A > G	Leu151Pro (7); Thr183Ile; Gln292Arg;	Two-component regulatory system PRRA/PRRB
		*moaC3*	392A > G	Asp131Gly	Probable molybdenum cofactor biosynthesis protein
		*iniB*	290C > T	Thr97Ile	Isoniazid inductible gene protein.
		*narL*	298G > C	Ala100Pro	Possible nitrate/nitrite response transcriptional regulatory protein
		*ctpl*	3113TGCGAG > T	Indels, frameshift variant	Probable cation-transporter ATPase I
296A	6	*TB18.5*	145G > C; 236A > G; 243C > A; 277A > G	Val49Leu (2); Tyr79Cys; His81Gln; Thr93Ala (2)	Conserved protein
		*Rv1948*	122C > A	Ala41Glu	Hypothetical protein
		*pks6*	667G > A	Val223Ile (2)	Probable membrane-bound polyketide synthase
648X	2	*ctpC*	1511C > T	Ser504Phe	Probable metal cation-transporting P-type ATPase C
454A	3	*Rv0678*	466 G > GC (3)	indels, frameshift variation	Conserved protein
		*mbtA*	1369CT > C (3)	Indels, frameshift variation	Bifunctional enzyme MbtA: salicyl-AMP ligase (SAL-AMP ligase) + salicyl-S-ArCP synthetase
1114A	2	*dnaE1*	2215A > G	Met739Val	Probable DNA polymerase III (alpha chain) DnaE1 (DNA nucleotidyltransferase)
		*virS*	983C > A	Pro328His	Virulence-regulating transcriptional regulator VirS (AraC/XylS family)
		*Rv0585c*	1202A > C	Asp401Ala	Conserved integral membrane protein
		*sugl*	16C > T	Gln6* (2), gained stop codon	Involved in transport of sugar across the membrane. Responsible for the translocation of the substrate across the membrane.
486X	3	*phoR*	661G > C	Ala221Pro,	Possible two-component system response sensor kinase membrane associated PhoR
		*fbiC*	1082C > A	Thr361Lys,	Probable F420 biosynthesis protein FbiC
		*fbiA*	866T > A	Leu289Gln (2)	Probable F420 biosynthesis protein FbiA
		*Rv3327*	296C > G	Pro100Ala	Probable transposase fusion protein
912A	3	*ethA*	205T > C; 190T > C	Trp69Arg; Phe64Ile	Monooxygenase that activates the pro-drug ETH
		*Rv2542*	1042G > A	Ala348Thr (2)	Conserved hypothetical protein
		*Rv3083*	783G > A; 806T > C	Trp261*, gained stop codon; Leu269Pro	Probable monooxygenase (hydroxylase)
622A	3	*ftsK*	1192T > C	Ser398Pro	Possible cell division transmembrane protein
		*virS*	712G > T	Val238Phe	Virulence-regulating transcriptional regulator VirS (AraC/XylS family)
		*sugl*	16C > T	Gln6* (3), gained stop codon	Involved in transport of sugar across the membrane. Responsible for the translocation of the substrate across the membrane.
705A	1	*Rv3083*	380_381 G > GA, indels, frame shift variant	Glu127_Thr128fs	Probable monooxygenase (hydroxylase)
		*sugl*	16C > T	Gln6*, gained stop codon	Involved in transport of sugar across the membrane. Responsible for the translocation of the substrate across the membrane.

## Data Availability

This whole genome shotgun project has been deposited in DDBJ/ENA/GenBank under BioProject PRJNA558545.

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
