# Peer review of "Hit Compounds and Associated Targets in Intracellular Mycobacterium tuberculosis"

_molecules, 2022, doi:10.3390/molecules27144446_

Round 1
Reviewer 1 Report
Solving the problem of drug resistance is one of the challenges facing modern phtisiology. Microorganism resistance to antibiotics is associated with mutations in the genome that lead either to a change in the target protein, or to increased expression of the gene encoding the target protein, the overproduction of which prevents the drug from blocking the target. Here the authors using a combination of in vitro breeding with whole genome sequencing anlysis were able to discover several new drug targets and drug resistance genes in Mtb. Thus, an important step towards development of new anti-TB drugs has been made.
Minor concern: Term "smile formulas" might be confusing to those not using PubChem Sketcher.
Author Response
Minor concern: Term "smile formulas" might be confusing to those not using PubChem Sketcher.
>> Accepted. Removed the term smile.
Author Response
Reviewer #2:
- Authors should properly summerized breifly the mechanisms of drug resistance in Mtb and include up to date literature review in the introduction section. Example: Line 29 which is an old information about TB burden worldwide
Thanks for this useful comment: Although we deal with hit compounds and not known drugs, we accepted this comment and referred the reader to recent review describing Mtb known mode of resistance.
We also updated the manuscript with most updated information
- Considering the significant variability among chemical compounds, authors need to explain how they have decided on the number and bases of the compounds used in this study?
We did not decide arbitrarily on the numbers. This is a result of over 5 years of accumulating data. We decided to release the information once a critical mass of data came.
- Results and Discussions of the manuscript could be shortened with a focus.
Thank you, we accepted this remark and shortened the manuscript as much as we could.
- I want to know if authors provide quality control metrics as a separate paragraphin
the methodology section.
We accepted this comment. We have added a sentence “Variants of phred score above 100 were selected for analysis”. Also, the variants were inspected individually . The phred score of variants is already included in the supplementary Table2.
- Please avoid citation of unpublished data from the text: line 94 & 98.
Accepted. The citation was removed
- The whole text should be improve dinthewritingstyles,theuseofwords,sentences
with clarity, rules of punctuation, grammar, spelling error, consistency with regards to abbreviations, and other simple formatting. Example: Line 233: CtpC/ ctpC, Line: 305: 1X 108 CFU/mL, Line: 309, CO2 and others through out the text
Read through and corrected
- Please indicate the exact value for multiplicity of infection(MOI)(line292)
Corrected
Reviewer 3 Report
Thank you for the opportunity to review this manuscript.
Overall, the finding presented in this manuscript is very interesting, which could contribute in the discovery of new drug/targeting agent towards MTB.
Several suggestions for consideration:
1. The write up is somehow rather confusing at certain parts of the manuscript. The authors may need to have a thorough check on this matter. Simplify the write up to help & ease understanding.
2. The title may be improved, by choosing more suitable term(s).
3. In the abstract, it was stated that "It causes chronic lung diseases to one third of the world’s 13 population". Please recheck this statement. The information in the Introduction differs: "One third of the world’s popula- 28 tion is exposed to Mtb".
4. Result & Discussion section can improved to enhance clarity.
5. What is/are the differences between Figure 1 and Figure 3? The font used in these figures are too small. Please improve the figures.
6. It is not clear what are the extracts and peptides mentioned in Section 3.3. Extracts are a mixture of compounds. But according to the write up, the authors used a library of compounds - I assumed they are pure compounds and not extract. Please elaborate more in the text.
Author Response
Reviewer #3
Several suggestions for consideration:
- The write up is somehow rather confusing at certain parts of the manuscript. The authors may need to have a thorough check on this matter. Simplify the write up to help & ease understanding.
We went through and attempted to simplify the write up.
- The title may be improved, by choosing more suitable term(s).
Accepted. The title was changed and simplified.
- In the abstract, it was stated that "It causes chronic lung diseases to one third of the world’s 13 population". Please recheck this statement. The information in the Introduction differs: "One third of the world’s popula- 28 tion is exposed to Mtb".
We thank the reviewer for noticing this obviously inaccurate statement. We corrected this issue. We apologize. The wrong statement was removed.
- Result & Discussion section can improved to enhance clarity.
The section has been revised.
- What is/are the differences between Figure 1 and Figure 3? The font used in these figures are too small. Please improve the figures.
The figures have been revised and improved.
- It is not clear what are the extracts and peptides mentioned in Section 3.3. Extracts are a mixture of compounds. But according to the write up, the authors used a library of compounds - I assumed they are pure compounds and not extract. Please elaborate more in the text.
Indeed, we checked compounds and not extracts. We accepted the comment and corrected the manuscript.